# CRISPR-induced double-strand breaks trigger recombination between homologous chromosome arms

Erich Brunner[1],* , Ryohei Yagi[2],* , Marc Debrunner[1] , Dezirae Beck-Schneider[1], Alexa Burger[1], Eliane Escher[1], Christian Mosimann[1], George Hausmann[1], Konrad Basler[1]

CRISPR–Cas9–based genome editing has transformed the life sciences, enabling virtually unlimited genetic manipulation of genomes: The RNA-guided Cas9 endonuclease cuts DNA at a specific target sequence and the resulting double-strand breaks are mended by one of the intrinsic cellular repair pathways. Imprecise double-strand repair will introduce random mutations such as indels or point mutations, whereas precise editing will restore or specifically edit the locus as mandated by an endogenous or exogenously provided template. Recent studies indicate that CRISPR-induced DNA cuts may also result in the exchange of genetic information between homologous chromosome arms. However, conclusive data of such recombination events in higher eukaryotes are lacking. Here, we show that in *Drosophila*, the detected Cas9-mediated editing events frequently resulted in germline-transmitted exchange of chromosome arms—often without indels. These findings demonstrate the feasibility of using the system for generating recombinants and also highlight an unforeseen risk of using CRISPR-Cas9 for therapeutic intervention.

## Introduction

CRISPR–Cas9–based genome editing has revolutionized genetic research, triggering the development of a plethora of technologies and applications that provide unprecedented control over genes in a growing list of model species (1, 2, 3, 4, 5, 6, 7, 8). CRISPR systems allow us to edit, engineer, or regulate genomes, hold great promise for clinical applications, and are likely to be used to treat diseases with genetic underpinnings, including cancer (9, 10). Genome editing is achieved by precisely targeting the nuclease activity of a modified bacterial protein (Cas9) via a user-defined guide RNA to a specific DNA sequence (1). The resulting DNA double-strand breaks (DSBs) are repaired either by the error-prone nonhomologous end joining (NHEJ) or homology-directed repair (11). For refined and precise genome editing purposes, homology-directed repair is harnessed to copy a specific DNA template (single-stranded or double-stranded) into the target site (2, 9, 12, 13). In contrast, NHEJ ligates the two broken ends of the DNA without a donor template, often resulting in random insertions or deletions (indels) that can disrupt coding sequences at the target site (for review see reference 14). However, with directly ligateable ends, NHEJ may lead to accurate repair of close and concurrent DSBs (15, 16, 17, 18), also when induced by Cas9 (19, 20, 21, 22). The ability of CRISPR–Cas9 to introduce several concurrent DSBs at defined positions has enabled engineering of tumor-associated chromosomal translocations resembling those observed in cancers, and hence to establish and test novel in vitro and in vivo tumor models (2, 23, 24, 25). Sadhu et al. (26) leveraged the CRISPR–Cas9 system to produce other chromosomal rearrangements, generating targeted mitotic recombination events in yeast to enable the fine mapping of trait variants. The authors deliberately induced a single DSB in one of the homologous chromosomes in a diploid yeast strain and achieved homologous recombination-based "loss of heterozygosity" events within 20 kb of the target site. Additional reports suggest that recombination in mitotic cells is not restricted to yeast but may also occur in other species such as houseflies (27) and tomatoes (28). What is currently missing is a solid confirmation of such events and data on their frequency in different species.

Here, we set out to examine the occurrence and frequency of genetic exchanges between homologous chromosome arms initiated by Cas9-induced DSBs. We show that Cas9-triggered DSBs induce germline-transmitted recombination between homologous chromosome arms in up to 39% of the CRISPR events in *Drosophila*. Although these findings expand the tool-box of CRISPR-based genome manipulation in research, they also raise concerns about the use of gene editing in therapeutic settings.

## Results

### CRISPR/Cas9 cuts induce recombination events

NHEJ is a major repair mechanism triggered by CRISPR–Cas9–induced DSBs in *Drosophila* (29, 30). Leveraging on this, we developed

---

[1]Institute of Molecular Life Sciences, University of Zurich, Zurich, Switzerland   [2]Institute of Molecular Systems Biology, Eidgenössische Technische Hochschule Zurich, Zurich, Switzerland

Correspondence: kb@imls.uzh.ch; erich.brunner@imls.uzh.ch
*Erich Brunner and Ryohei Yagi contributed equally to this work

a system for activating transgene expression through NHEJ-based repair. The system, which we named CIGAR (CRISPR-Induced Gene Activator), allows activation of transgene expression after CRISPR-induced DSBs. The principle of the CIGAR system is based on activation of gene expression if, and only if, a unique CRISPR–Cas9 target sequence has been cleaved and rearranged by NHEJ (Fig 1). CIGAR consists of four elements: (i) the *ubiquitin-p63E* promoter to drive gene expression in every cell (31), (ii) a so-called "shifter" sequence, (iii) a flexible linker sequence inserted 3′ of the shifter sequence (32), and (iv) a reporter cDNA (lacking a translational start codon) followed by the *3′UTR* of the *Drosophila tubulin α1* gene. The functional core of the *CIGAR* reporter lies within the shifter sequence, which contains optimized translational *START* codons covering all three frames upstream of a unique 20-nt CRISPR target sequence (33, 34). Each initiation codon is blocked downstream by a corresponding in-frame *STOP* codon. Importantly, the most 5′ *STOP* codon, named *STOP^T* (*T* for *target*), is in-frame with the downstream ORF and resides within the unique 20-nt *gRNA* target, starting 4 nt upstream of the *PAM* sequence. Activation of *CIGAR* is achieved by Cas9-induced DNA cleavage within the *STOP^T* codon. The induced DSBs will then be mended by NHEJ-mediated repair concomitantly severing or eliminating the *STOP^T* codon. The resulting indel leads to repositioning of the upstream *ATGs* relative to the ORF, that is, causing one of the *ATGs* to be "shifted in-frame" with the ORF.

Two variants of this CIGAR tool box, that is, *CIGAR^eGFP* or *CIGAR^mCherry* are depicted in Figs 1 and 2A. Further details of the CIGAR system will be submitted elsewhere. When we molecularly analyzed the shifter sequences in the progeny of females carrying the two

*CIGAR* transgenes *in trans* (genotype: *nos-Cas9, CIGAR^eGFP/CIGAR^mCherry; U6:3-sgRNA^CIGAR(1,2)/+*), we found that in some of the F1 animals, the CRISPR target sequence of the *CIGAR^mCherry* reporter became located 5′ of the *eGFP* ORF and vice versa. Sequence analysis of 84 animals from different crosses revealed a total of 26 animals in which the sequences on one side of the DSB had been exchanged (Figs 2B, S1, and S2 and Table S1). The site of the DSB was in part marked by indels. From these results, we concluded that in these animals, recombination events had occurred at the site of the Cas9-induced DSBs. We would like to emphasize that the term recombination is used here to describe the exchange of genetic material between homologous chromosome arms initiated by CRISPR-induced DSBs. Because Cas9 activity may often cause the break of both homologs, such recombination events may not only be based on homologous recombination but may also result by breakage/fusion events involving NHEJ that lead to a crosswise ligation of the chromosome arms (15, 16, 17, 18, 19, 20, 21, 22). From our results, however, we cannot infer which repair mechanism was involved in the exchange of genetic information between homologous chromosome arms.

In the experiment described above, the CIGAR transgenes were located at position 5D on the X-chromosome that shows native recombination activity in the female germline (35). To further test if CRISPR–Cas9 could induce efficient site-specific recombination, we turned to the fourth chromosome for which normally no naturally occurring recombination is observed (36, 37, 38). We first generated transgenic animals harboring the *CIGAR^eGFP* and *CIGAR^mCherry* construct at position *102F* on the two homologous arms of chromosome 4, in addition to *nos-Cas9 and U6:3-sgRNA^CIGAR(1,2)*. Such

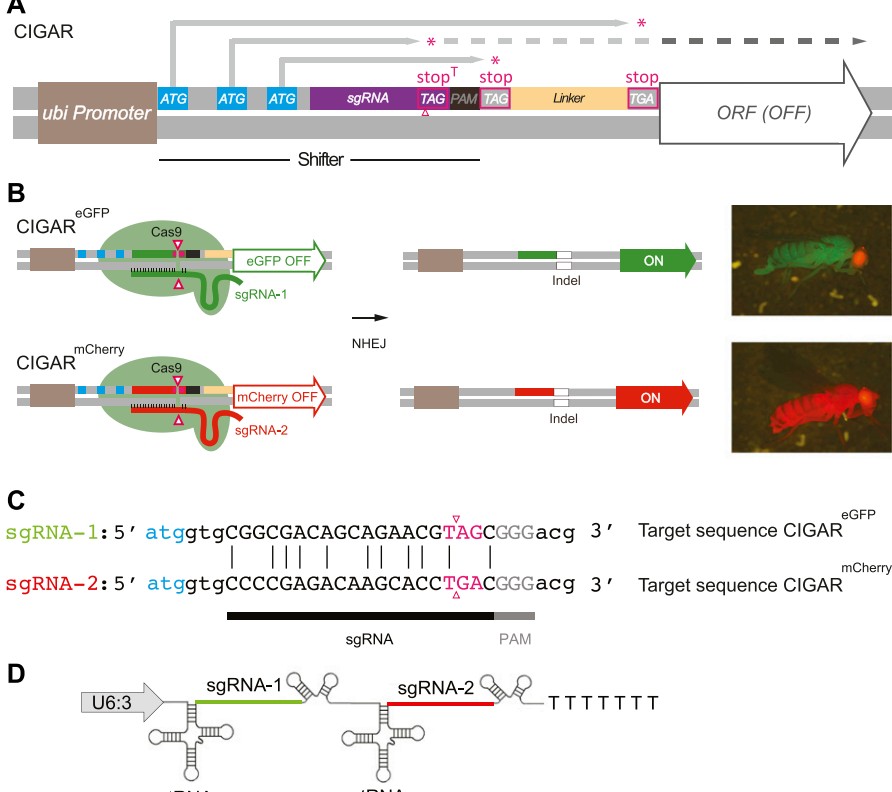

**Figure 1.  *CIGAR* design and activation.**
**(A)** *CIGAR* consists of four elements: 1. Ubiquitin-p63E promoter (brown box), 2. "shifter" sequence, 3. linker sequence (orange), 4. reporter cDNA (ORF) lacking a translational start site followed by the tubulin 3′UTR (grey open arrow). The shifter sequence contains optimized translation initiation codons covering all three frames (light blue boxes) and a guide RNA target region (purple box) followed by a protospacer adjacent motif (PAM) (black box). In the inactive *CIGAR*, translation from each ATG (grey arrows) is terminated 5′ of the ORF by a *STOP* codon (red asterisks) preventing the translation of the downstream ORF. **(B)** Activation of two *CIGAR* variants harboring either an *eGFP* or a monomeric *Cherry* (*mCherry*). Activation is achieved by Cas9-induced DNA cleavage within stop^T (pink), the most upstream *STOP* codon which is in-frame with the downstream ORF. The resulting double-strand break (DSB; the putative site of the DSB, 3 bp upstream of the PAM site is indicated with a pink, open arrowhead) is mended by NHEJ-mediated repair concomitantly eliminating stop^T (indel; open white box) and shifting one of the ATGs in-frame with the ORF. Flies that inherited a translationally activated *CIGAR* appear uniformly green or red. The eye-specific red fluorescence marks the attP target site into which the *CIGAR* constructs have been inserted. **(C)** Comparison of the 20-nt *sgRNA* target sequences of *CIGAR^eGFP* and *CIGAR^mCherry*, respectively. The 9-bp substitutions in the two target sites are indicated. **(D)** For specific and simultaneous targeting of both reporters, a *tRNA*-spaced tandem array (*U6:3-sgRNA^CIGAR(1,2)* harboring *sgRNA-1* (targeting *CIGAR^eGFP*) and *sgRNA-2* (targeting *CIGAR^mCherry*) is used.

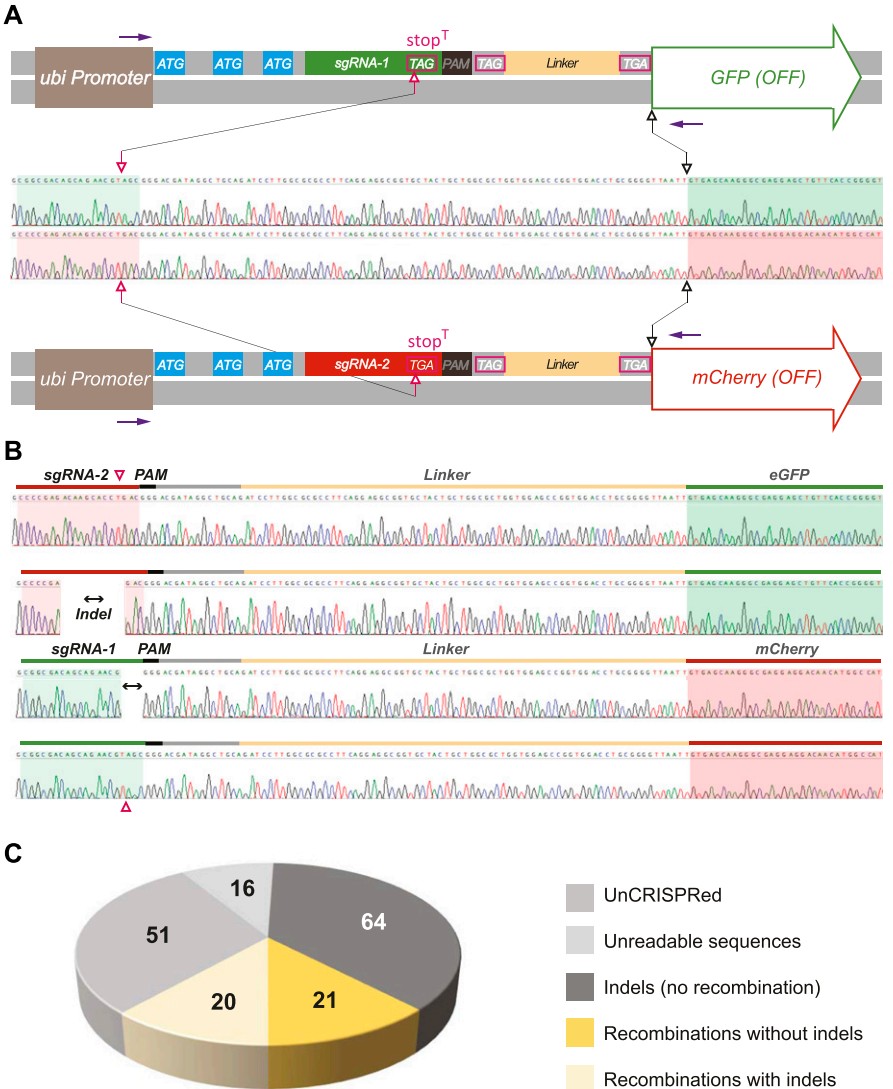

**Figure 2. Detailed *CIGAR^eGFP* and *CIGAR^mCherry* reporter design and illustration of recombination events on the sequence level.**
**(A)** Design and sequence details of un-CRISPRed *CIGAR^eGFP* (top) and *CIGAR^mCherry* reporters (bottom). The sequences of the *sgRNAs* and the ORFs are shaded in green and red, respectively. Note that except for the *sgRNAs* and the ORFs, the sequences of the reporters are identical. The targeted STOP codon (stop^T; pink) differ in sequence. The CRISPR target sites are delineated in the sequence context (pink, open arrows). Analysis of the shifter sequence is performed using primer pairs specific for the Ubi promoter and the 5′ end of the respective ORF (purple arrows). **(B)** The shifter region of flies harboring a single copy of one of the *CIGAR* reporters on the X chromosome (attP 5D) was analyzed by single fly PCR and Sanger sequencing. Shown are recombination events from a *CIGAR^eGFP*/*CIGAR^mCherry* co-targeting experiment visualized on the sequence level. As in (A), the sequences of the *sgRNAs* and the ORFs are shaded in green and red, respectively. Recombinants exhibit a rearranged sequential arrangement (green-red or red-green) of *sgRNA* and reporter cDNA. Note that recombination events may or may not be accompanied by indels at the target site. **(C)** Co-targeting experiments using *CIGAR* reporters on the fourth chromosome (attP 102F). 172 animals were analyzed by single fly PCR and Sanger sequencing. The yellow sections represent the number of recombinants with or without indel.

individuals were crossed to *yw* animals, and their offspring were scored for recombination events (Fig 2C, see the Materials and Methods section for details). Animals were randomly picked from 13 crosses immediately after hatching and analyzed for the sequences flanking the target site (Tables S2 and S3). This revealed that in 41 of 156 animals, recombination events had occurred (Fig 2B; for details see the Materials and Methods section). Thus, as described above for the X chromosome, numerous CRISPR-mediated recombination events were observed on the fourth chromosome (39% of the detected Cas9-triggered events, i.e., 41 of 105 CRISPR events).

### CRISPR-induced recombination between two distant phenotypic markers

In the above experiments, recombination was induced between homologous chromosome arms for which the nt at the Cas9 target site as well as the flanking sequences (coding for the fluorescent proteins) differed. However, the recombination events could only

be demonstrated by sequence analysis of the immediate vicinity of the CRISPR site; more distant phenotypic markers were not present on these chromosomes. Therefore, we could not rule out that, at least in some cases, other mechanisms, such as gene conversion, were responsible for the observed sequence exchange between the two CIGAR reporters *in trans* (39).

To confirm that indeed a complete exchange of the homologous chromosome arms occurs distal to the CRISPR–Cas9–induced DSBs, we used two visible markers separated by more than 100 kb (Figs 3A and S3): the *w^+*-marked *CIGAR^mCherry,102F, w+* and the recessive viable mutation *sv^spa-pol* close to the tip of the right arm on chromosome 4. We selected the Cas9 target site, targeted with *sgRNA-3*, in the *3′UTR* of the *toy* gene residing about 18 kb downstream of the *CIGAR^mCherry,102F, w+* transgene insertion site. To induce recombination at the target site (Fig 3A, TR; see the Materials and Methods section for more experimental details), we injected *yw; CIGAR^mCherry,102F,w+*/*Dp(1;4)1021,y^+*, *sv^spa-pol* embryos with active Cas9-sgRNA RNP complexes containing recombinant Cas9 and in vitro–translated *sgRNA-3* (Fig 3B) (40, 41). *G0*

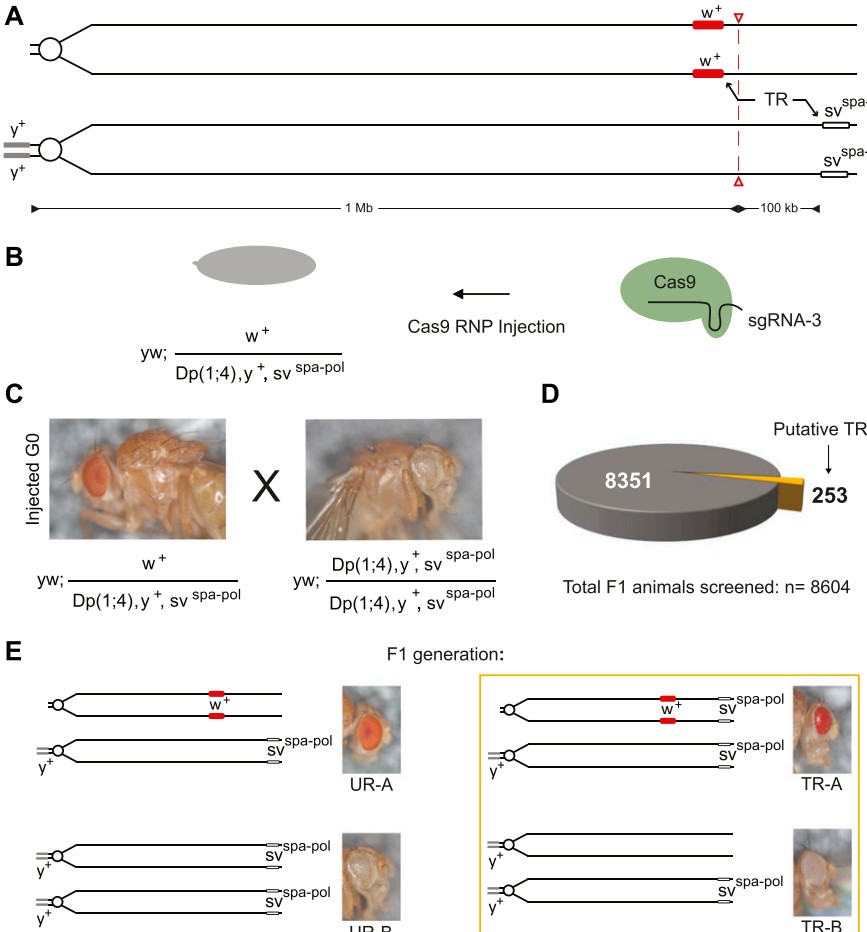

**Figure 3. Cas9-induced recombination between two phenotypic markers on the fourth chromosome.**
**(A)** Each of the two markers ($w^+$ [*mini-white*] and $sv^{spa-pol}$) is located on a different homologous chromosome. The two markers are separated by about 100 kb. The $sv^{spa-pol}$ chromosome is marked with $y^+$ due to a duplication of X-chromosomal material to the short, left arm of chromosome four. The Cas9 cut site is represented by a red dashed line and red open arrow heads. The repair of CRISPR-induced DSB may lead to targeted recombination events (TR) between the two markers. **(B)** Embryos with the genotype *yw; CIGAR$^{mCherry,102F,w+}$/ Dp(1;4)1021,y$^+$, sv$^{spa-pol}$* were injected with recombinant Cas9 RNPs containing in vitro–translated *sgRNA-3*. **(C)** Cas9 RNP injected G0 animals are backcrossed to animals with the genotype *yw; Dp(1;4)1021,y$^+$, sv$^{spa-pol}$/ Dp(1;4)1021,y$^+$, sv$^{spa-pol}$* to be able to visually score putative recombinants. The phenotype of the animals is shown. **(D)** A total of 8,604 animals were screened and 253 putative recombinants were recovered. **(E)** Unrecombined animals (UR) appeared phenotypically as $y^+$; $w^+$ (UR-A) or $y^+$; $sv^{spa-pol}$ (UR-B). Putative recombinants (boxed) presented either as $y^+$; $w^+$; $sv^{spa-pol}$ (TR-A) or $y^+$, $sv^+$ animals (TR-B).

animals were crossed with homozygous *yw; Dp(1;4)1021,y$^+$, sv$^{spa-pol}$* flies to visually detect recombination events in the offspring (i.e., the two visible markers co-segregate upon recombination; see Fig 3B and C and see the Materials and Methods section for details). From 8,604 offspring, we recovered 253 putative recombinants (Fig 3D). Of these 253 animals, we further characterized 57 deriving mostly from independent crosses. 21 of the 57 animals turned out to be true germline-transmitted recombinants that were homozygous viable (TR-A and TR-B in Fig 3E and Tables S4 and S5). Molecular analysis of the Cas9 target site revealed that 19 of these 21 recombinants did not contain an indel lesion at the CRISPR site (sequence traces with indels are shown in Table S4). Extrapolating from the 21 confirmed recombinants (out of 57) to the 253 *putative* recombinants (out of 8,604 F1 animals), the frequency of CRISPR-mediated targeted recombination in this experiment is estimated to be ~1.1%. This frequency is at least four orders of magnitude higher than that of the rare spontaneous recombination rate predicted for chromosome 4 (37).

## CRISPR-induced DSBs may lead to loss of chromosomal structures

The remaining 36 (out of 57) putative recombinants recovered from the above experiment were homozygous lethal and exhibited position effect variegation (PEV; Fig 4A and B) in the adult compound eye. PEV

in the eye occurs if the *mini-white* gene ($w^+$), used as transgene reporter, is juxtaposed to heterochromatic regions via chromosomal rearrangements or translocations (42, 43, 44, 45). In particular, the proximity of the *mini-white* gene to the heterochromatic telomere regions may lead to PEV (46). We, therefore, reasoned that in animals showing PEV, the Cas9-based editing led to a loss of chromosome structures distal to the DSB and consequently, expression of the *mini-white* gene of the CIGAR transgene is variably silenced. Similar events have been reported for X-ray–induced DSBs on the fourth chromosome (47). To assess the presence or absence of the most distal part of the chromosomes showing PEV, we tested *CIGAR$^{mCherry,102F,w+}$/* chromosomes for complementation of the lethal $sv^{Δ122}$ null mutation. Independently recovered *yw; CIGAR$^{mCherry,102F,w+,PEV}$/Dp(1;4)1021,y$^+$, sv$^{spa-pol}$* animals exhibiting PEV were crossed to *yw; ci$^D$,sv$^{spa-pol}$/sv$^{Δ122}$* animals. Without exception, the tested PEV chromosomes were unable to complement the $sv^{Δ122}$ mutation, indicating that they must have lost distal parts of chromosome 4.

## Recombination is triggered by, and confined to the site of, Cas9-induced DSBs

Finally, to exclude that Cas9-induced DSBs merely stimulate nonspecific recombination (NSR) on the fourth chromosome (i.e.,

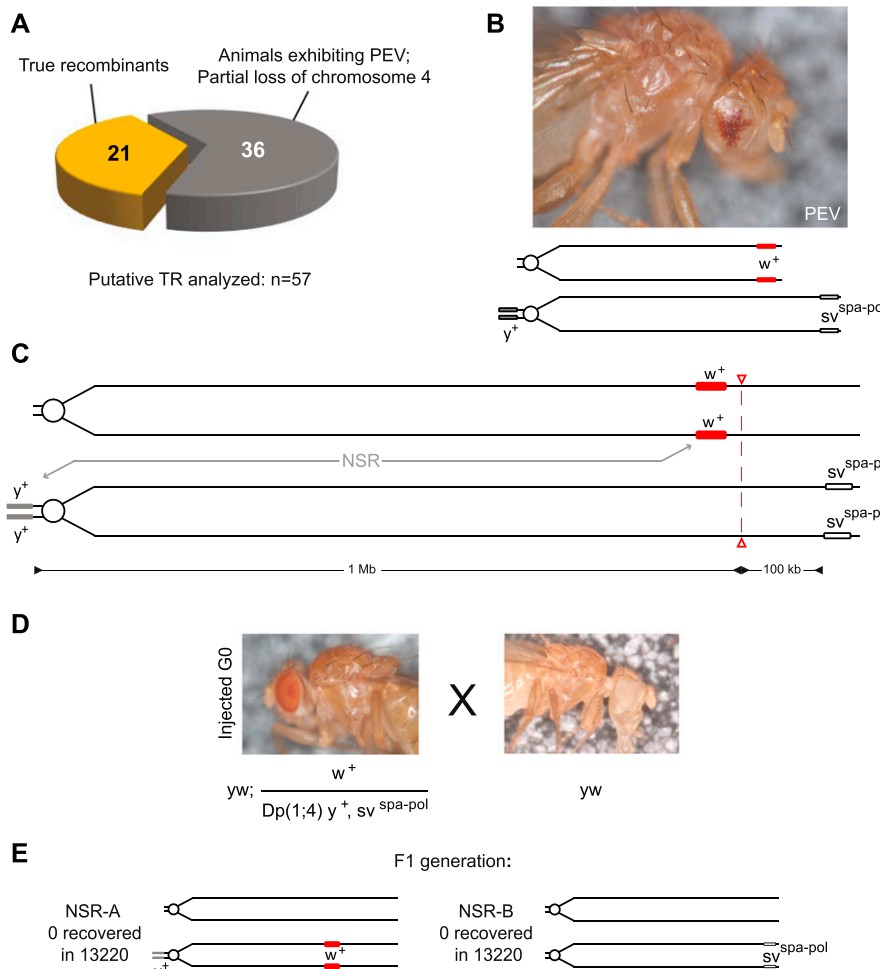

**Figure 4.   CRISPR-induced DSBs may lead to loss of chromosomal structures.**
**(A)** Of the putative 257 recombinants (see Fig 3D), 57 were characterized in more detail. 21 were true recombinants, whereas 36 animals had a loss of chromosomal structures distal to the CRISPR target site which manifested in PEV, that is, variable expression of the mini-white gene. **(B)** Fly exhibiting PEV is shown. The genotype of the fly and the loss of chromosomal structures distal to the cut site are shown below the image. Important to note is that only the loss of distal chromosome structures of the $w^+$ chromosome can be scored from the TR screen (described in Fig 3). The corresponding loss of $sv^{spa-pol}$ on the $Dp(1;4)1021,y^+$ chromosome is phenotypically identical to UR-B (Fig 3E) and will not be recovered from the screen. **(C)** Control experiment to assess if CRISPR–Cas9–induced DSBs in general enhances the frequency of recombination away from the Cas9 cut site (dashed line and red open arrowheads). **(D)** Cas9 RNP–injected animals (same as in Fig 3B) are backcrossed to animals with the genotype $yw$ to be able to visually score NSR events between $w^+$ and $y^+$. **(E)** No NSR between $y^+$ and $w^+$ marker (NSR-A and NSR-B) were recovered amongst 13,220 animals screened (see text and the Materials and Methods section for more details).

not at the Cas9 target site), we repeated the experiment using the same experimental conditions, but assessed recombination between a $y^+$ marker on the short left arm of chromosome 4 and the $w^+$-marked $CIGAR^{mCherry,102F}$ transgene inserted at *102F* of the right arm (Fig 4C–E, NSR; see the Materials and Methods section for more experimental details). Embryos of the genotype $yw; CIGAR^{mCherry,102F, w+}$ / $Dp(1;4)1021, y^+, sv^{spa-pol}$ were injected with RNPs containing Cas9 protein complexed with in vitro–translated *sgRNA-3*. The RNPs induce DSBs *3′* of *toy* located distal of $CIGAR^{mCherry,102F}$ but should not have any influence on the recombination between the $y^+$ and the $w^+$ markers. G0 animals were crossed with $yw$ animals to score recombination events between the $y^+$ and the $w^+$ marker (Fig 4D). We screened more than 13,000 F1 animals but did not observe a single recombination event between the markers (NSR-A or NSR-B in Fig 4E, see the Materials and Methods section for details). These results are consistent with the notion that no spontaneous recombination occurs between the fourth chromosomes and that the recombination observed in the previous experiments above (Fig 3, TR) were triggered by, and confined to the site of, CRISPR-induced DSBs.

In summary, our experiments involving chromosome 4 revealed that germ-line–transmitted recombinants can be recovered at frequencies ranging from about 1.1% (recombination between

visible markers) to 26% (recombination between CIGAR constructs; see the Materials and Methods section). These percentages represent the number of recombinants recovered from the total number of animals analyzed. When only the detected Cas9-mediated events are taken into consideration, the frequency of recombination was even 39%.

## Discussion

Our results show that a substantial amount of CRISPR–Cas9–induced DSBs result in exchanges between homologous chromosome arms. Importantly, the recombination events we see occur in multiple experimental settings and not only under specific conditions. In this context, it is important to point out that recombination events observed in experiments where Cas9 has been provided as recombinant protein exclusively occurred in mitotic and not in meiotic cells: a study by Burger and colleagues shows that fluorescently labeled Cas9 RNP complexes are detectable until about 18 h after injection (40). In *Drosophila,* however, the first meiotic divisions occur at much later timepoints (i.e., in males in the

third instar larvae, about 3–4 d after injection and in females at early pupal stages ([48])), likely excluding residual activity of Cas9 RNPs.

In experiments where Cas9 protein and the *sgRNAs* are provided via transgenes ([49], [50]), Cas9 expression is driven by the *nanos (nos)* promoter and the Cas9 transgene contains the *nos* 3′UTR recapitulating germline-specific *nos* expression, transcript localization, and translational control ([49], [51], [52]). Hence, Cas9 is maternally provided to the offspring and likely not expressed zygotically during embryogenesis ([53]). In females, Nos is available during mitotic divisions in the germline stem cells as well as in those that later on form the 16-cell cyst in the germline. It should not be present in the growing oocyte where meiosis would occur ([53]). In the male germline, *nos* expression seems to be essential during spermatogenesis ([54]). Loss of *nos* expression leads to various phenotypes with strongest effects on the number of primary spermatocytes that are created through mitotic divisions. Together, these arguments suggest that also in case the CRISPR reagents are provided via transgenes, recombination events occurred preferentially in mitotic cells. However, because the *nos-Cas9* transgene is not inserted into the *nos* locus, presence of Cas9 protein and extension recombination events in meiotic cells cannot strictly be ruled out.

One reason why CRISPR-induced recombination has largely remained unnoticed in genetic model organisms is that DSBs are frequently repaired without resulting indels and thus cannot be detected by next-generation sequencing, preventing any follow-up validation of broader effects. When detected using genetic and sequencing validation, we noted variable recombination rates in our experiments. We attribute this variability to three experimental aspects. First, the different Cas9 target sites may have different cutting and recombination efficacy ([30], [55]). Second, individual crosses may yield different numbers of progeny harboring CRISPR-derived indels (i.e., CRISPRed (G0) animals transmit a variable number of mutant alleles to the next generation, ranging from 0 to 100% as shown for the recovery of a nonfunctional *y* allele, see Fig 2 in reference [49]). Third, in our experiments, different methods were used to introduce Cas9 and *gRNAs*, a circumstance that likely contributed to the different frequencies of recombinants recovered ([49]).

The Cas9-mediated site-specific chromosomal recombination described here opens up a number of new avenues and considerations for genome engineering. We demonstrate that in *Drosophila*, Cas9-mediated DSBs can be used to generate recombination at a predefined site between chromosomal locations that are in close vicinity. Unlike for site-specific recombination based on Flp ([56], [57]), no recombinase target sites need to be present. Thus, CRISPR-mediated site-specific recombination enables the combination of two known mutations situated in different alleles of one and the same gene into a single (double-)mutant allele. Moreover, it could be used to study permutations of mutant alleles in tightly linked genes such as members of a *Hox* gene cluster ([58]). Especially in vertebrates, such as mouse or zebrafish, CRISPR-induced recombination holds promise for a number of applications, including generation of complex mutant alleles in the same locus ([59]). Also, in *Drosophila*, Cas9-mediated recombination enables experiments that were previously impossible because of low or absent recombination rates. For example, by providing *sgRNAs* or Cas9 activity in a tissue-specific manner, it may become possible to generate specific mutant clones for genes located on the fourth chromosome. Finally, targeted Cas9-mediated recombination could potentially be used in epigenetic studies to determine the effects of swapping promoters, including their epigenetic marks, between maternal and paternal genes.

On the other hand, our findings that recombination between homologous chromosome arms may be triggered upon Cas9-induced DSBs underscore the need for caution in applying CRISPR-based genetic interventions in animals or humans ([60]): Unrecognized CRISPR-induced recombination events (i.e., no visible indels at the target site) may lead to loss-of-heterozygosity events, generating cells with unnoticed homozygosity for imprinted genes or recessive mutations located distal to the Cas9 target site, which may have unforeseen consequences.

Moreover, we observed that CRISPR-based engineering can lead to loss of chromosome material distal to the locus of the DSB (Fig 4B). We could observe such events in our system because haplo-four animals (harboring only a single copy of the fourth chromosome) are viable ([61]). Under most circumstances, and likely in most organisms, such events would lead to the death of the affected cells because of haploinsufficiency. In this context, recent work suggested that Cas9 induces mutations in human cells and mice that are larger than anticipated ([62]). Therefore, it is imperative to routinely consider broader chromosomal alternations as possible outcome when applying CRISPR technologies in translational medicine.

# Materials and Methods

### Plasmid construction

Unless otherwise noted, the plasmids were constructed by standard molecular cloning methods. When plasmids contain newly synthesized nt sequences via PCR, oligonucleotide synthesis, or mutagenesis, the sequences were verified by DNA sequencing.

### *pUbiattB*

The *StuI-ubiquitin p63E* promoter (*ubi*)-*EcoRI* fragment from *pCaSpR3-Up2-RX polyA* was subcloned into a *pBluescript* (*pBS*) vector between the *EcoRI* and *XhoI* sites using blunt-end ligation. The *EcoRI-ubi-Acc65I* fragment from the resulting plasmid was subcloned into *pEPattB* ([63]) using the *EcoRI* and *Acc65I* sites.

### pUbiattB-CIGAR^eGFP

To create the *pUbiattB-CIGAR^eGFP* reporter, the shifter sequence (containing optimized translational START codons covering all three frames upstream of a unique 20-nt CRISPR target sequence), a unique *gRNA* target sequence (referred to as *sgRNA-1*), the linker sequence, and the *eGFP* gene were designed (as shown below), synthesized by GenScript, and delivered ligated into the *pUC57-Kan* vector (*pCIGAR-D0*).

The *pCIGAR-D0* insert:

*5′-KpnI_CAACATGGTGCAACATGGTGCAACATGGTGCGGCGACAGCAGA
ACGTAGCGGGACGATAGGCTGCAGATCCTTGGCGCGCCTTCAGGAGGCGGT-*

GCTACTGCTGGCGCTGGTGGAGCCGGTGGACCTGCGGGGTTAATTGTGAGC-
AAGGGCGAGGAGCTGTTCACCGGGGTGGTGCCCATCCTGGTCGAGCTGGAC-
GGCGACGTAAACGGCCATAAGTTCAGCGTGTCCGGCGAGGGCGAGGGCGA-
TGCCACCTACGGCAAGCTGACCCTGAAGTTCATCTGCACCACCGGCAAGC-
TGCCCGTGCCCTGGCCCACCCTCGTGACCACCCTGACCTACGGCGTGCAGT-
GCTTCAGCCGCTACCCCGACCACATGAAGCAGCACGACTTCTTCAAGTCCGC-
CATGCCCGAAGGCTACGTCCAGGAGCGCACCATCTTCTTCAAGGACGACGG-
CAACTACAAGACCCGCGCCGAGGTGAAGTTCGAGGGCGACACCCTGGTGAA-
CCGCATCGAGCTGAAGGGCATCGACTTCAAGGAGGACGGCAACATCCTG-
GGGCACAAGCTGGAGTACAACTACAACAGCCACAACGTCTATATCATGGCC-
GACAAGCAGAAGAACGGCATCAAGGTGAACTTCAAGATCCGCCACAACATC-
GAGGACGGCAGCGTGCAGCTCGCCGACCACTACCAGCAGAACACCCCCATC-
GGCGACGGCCCCGTGCTGCTGCCCGACAACCACTACCTGAGCACCCAGTC-
CGCCCTGAGCAAAGACCCCAACGAGAAGCGCGATCACATGGTCCTGCTG-
GAGTTCGTGACCGCCGCCGGGATCACTCTCGGCATGGACGAGCTGTACAAG-
TAAGAATTC_EcoRI-3′.

The synthetic sequence was excised with *KpnI* and *EcoRI* and
ligated into pre–double-digested *pKB342* vector in line with a *tu-
bulin* 3′ trailer, transformed, and purified, resulting in *pKB342_CI-
GAR*. To remove the CIGAR sequence including the *tubulin* trailer,
*pKB342_CIGAR* was then digested with *KpnI* and *XbaI*, and the
fragment was ligated into a *KpnI*- and *XbaI*-digested *pUbiattB*
vector containing an *ubiquitin-p63E* promoter (see above). This
resulted in the final product referred to as *pUbiattB-CIGAR*<sup>eGFP</sup>.

## pUbiattB-CIGAR<sup>mCherry</sup>

The same sequential digestions and ligation as for the making of the
*pUbiattB-CIGAR*<sup>eGFP</sup> were used for the construction of the *pUbiattB-
CIGAR*<sup>mCherry</sup>. The only difference was in the design of the target site;
the *gRNA* (*sgRNA-2*) has a different unique target sequence and
contains the *mCherry* gene as the fluorescent marker. The insert
ligated into the *pUC57-Kan* vector was ordered from *GenScript*.

The *CIGAR*<sup>mCherry</sup> insert:

5′-KpnI_CAACATGGTGCAACATGGTGCAACATGGTGCCCCGAGCAAGC-
ACCTGACGGGACGATAGGCTGCAGATCCTTGGCGCGCCTTCAGGAGGCGGT-
GCTACTGCTGGCGCTGGTGGAGCCGGTGGACCTGCGGGGTTAATTGTGAG-
CAAGGGCGAGGAGGACAACATGGCCATCATCAAGGAGTTCATGCGCTTTAA-
GGTGCACATGGAGGGCTCCGTGAACGGCCACGAGTTCGAGATCGAGGGC-
GAGGGCGAGGGCCGCCCCTACGAGGGCACCCAGACCGCCAAGCTGAAGG-
TGACCAAGGGCGGCCCCCTGCCCTTCGCCTGGGACATCCTGTCCCCTCAGTT-
CATGTACGGCTCCAAGGCCTACGTGAAGCACCCCGCCGACATCCCCGACTAC-
TTGAAGCTGTCCTTCCCCGAGGGCTTCAAGTGGGAGCGCGTGATGAACTTCG-
AGGACGGCGGCGTGGTGACCGTGACCCAGGACTCCTCCCTGCAGGACGGCG-
AGTTCATCTACAAGGTGAAGCTGCGCGGCACCAACTTCCCCTCCGACGGCCCC-
CGTAATGCAGAAGAAGACCATGGGCTGGGAGGCCTCCTCCGAGCGGATGTA-
CCCCGAGGACGGCGCCCTGAAGGGCGAGATCAAGCAGAGGCTGAAGCTGAA-
GGACGGCGGCCACTACGACGCCGAGGTCAAGACCACCTACAAGGCCAAGAA-
GCCCGTGCAGCTGCCCGGCGCCTACAACGTCAACATCAAGCTGGACATCACC-
TCCCACAACGAGGACTACACCATCGTGGAACAGTACGAGCGCGCCGAGGGC-
CGCCACTCCACCGGCGGCATGGACGAGCTGTACAAGTAA_EcoRI-3′.

## U6:3-sgRNA<sup>CIGAR(1,2)</sup> (*pCFD5-F1*)

The *pCFD5* vector was a gift from Fillip Port (#73914; Addgene) (50).
*pCFD5* is the main backbone containing the tRNA assembly pre-
pared to insert multiple *gRNAs*. For our purposes, *sgRNA-1* and
*sgRNA-2* were inserted via Gibson cloning into the *pCFD5* vector

(named *pCFD5-F1*) following the protocol described in the sup-
plementary methods of *pCFD5* cloning protocol.

*pCFD5* internal tRNA multi-*gRNA* Scaffold:

5′-GTCGGGGCTTTGAGTGTGTGTAGACATCAAGCATCGGTGGTTCAGTGG-
TAGAATGCTCGCCTGCCACGCGGGCGGCCCGGGTTCGATTCCCGGCCGATG-
CAGGGTCTTCGTTTTAGAGCTAGAAATAGCAAGTTAAAATAAGGCTAGTCCGT-
TATCAACTTGAAAAAGTGGCACCGAGTCGGTGCAACAAAGCACCAGTGGTC-
TAGTGGTAGAATAGTACCCTGCCACGGTACAGACCCGGGTTCGATTC-
CCGGCTGGTGCAGAAGACCTGTTTTAGAGCTAGAAATAGCAAGTTAAAATAAG-
GCTAGTCCGTTATCAACTTGAAAAAGTGGCACCGAGTCGGTGCTTTTTT-3′.

Through the process of Gibson cloning, both our *sgRNA-1* and
*sgRNA-2* were inserted into the *pCFD5* vector using primer (5′-
GCGGCCCGGGTTCGATTCCCGGCCGATGCACGGCGACAGCAGAACGTA-
GCGTTTTAGAGCTAGAAATAGCAAG-3′) for *sgRNA-1* and (5′-ATTT-
TAACTTGCTATTTCTAGCTCTAAAACGTCAGGTGCTTGTCTCGGGGTGCAC-
CAGCCGGGAATCGAACCC-3′) for *sgRNA-2*.

## IVT of *sgRNA-3*

IVT of *sgRNAs* were performed as described in reference 40. For IVT,
we used MEGAscript (AM1334), and for the purification of the IVT
products, we used Purification: MEGAclear (AM1908).

IVT oligo used specific for *sgRNA-3*:

5′-GAAATTAATACGACTCACTATAGG<u>CTGTTGATAAGCACGCAATC</u>**GTTT-
TAGAGCTAGAAATAGC**-3′.

IVT oligo used *sgRNA-R* used for template PCR:

5′-AAAAGCACCGACTCGGTGCCACTTTTTCAAGTTGATAACGGACTAGCC-
TTATTTTAACTT**GCTATTTCTAGCTCTAAAAC**-3′.

Complementary sequences of the specific *sgRNA* primer and the
*sgRNA-R* are shown in bold letters.

## Cas9/*sgRNA* RNP injections

The concentration of SpCas9 injected was about 800 ng/µl SpCas9
(final concentration) and about 300 ng/µl of *sgRNA* (IVT; final
concentration after purification). This corresponds roughly to a 1:2
ratio (SpCas9*: sgRNA-3)*. SpCas9 has about 5× the molecular weight
of the *sgRNA*.

The injection mix is prepared as follows (total volume of 10 µl):
add *sgRNA*, ddH$_2$O, and 10× incubation buffer NEB and mix thor-
oughly. Gently add the SpCas9 and mix again thoroughly by
pipetting up and down. It is mandatory to add SpCas9 as the last
ingredient because low salt concentrations may cause SpCas9 to
precipitate (40).

X µl *sgRNA* (IVT; to a final concentration of 320 ng/µl; corresponds
to 1:2 ratio *Sp*Cas9*: sgRNA*).

1 µl 10× NEB incubation buffer.

Y µl ddH$_2$O (RNase free) to a total of 10 µl (X + Y = 6.5 µl).

2.5 µl SpCas9 (EnGen Cas9 NLS, NEB #M0646T; 3.22 µg; final
concentration 0.8 µg/µl).

Mix gently and incubate the mix at 37°C for 2 min.

Load the mix onto a column (Ultrafree-MC-HV 0.45 µm [Ref:
UFC30HV00]).

Spin for 1 min in a table-top centrifuge @14,000*g*.

Reincubate the flow-through at 37°C for 2 min.

Let the mix equilibrate at RT for ~30 min before injection. Never
put the mix back on ice.

### Fly genetics

Crosses were done at 25°C. Unless noted otherwise, fly lines were obtained from the Bloomington *Drosophila* Stock Center (see the Acknowledgments section).

Transgenic CIGAR fly lines (integration into *ZH-attP 5D* [X chromosome] or *ZH-attP 102F* [fourth chromosome]) were generated by *phiC31* integrase-mediated transgenesis ([51], [64]). Individual strains were confirmed to carry the correct shifter sequence by sequencing the PCR product of the shifter sequence using primers *CIGAR-fwd*: CAACAAAGTTGGCGTCGATA and *CIGAR^eGFP^-rev*: GAACTTCAGGGTCAGCTTGC (452 bp; for *CIGAR^eGFP^*); *CIGAR-fwd*: CAACAAAGTTGGCGTCGATA and *CIGAR^mCherry^-rev*: AAGCGCATGAACTCCTTGATG (367 bp; for *CIGAR^mCherry^*), respectively. PCR settings were as follows: 95°C, 5 min; 35 cycles of 95°C, 25 s; 60°C, 25 s; and 72°C, 30 s); final elongation of 72°C, 10 s. The same PCR setting was used to analyze the shifter sequence of the CIGAR reporters by single fly PCR.

The *pCFD5-F1* containing *U6:3-sgRNA^CIGAR(1,2)^* was inserted into the *attP 40* site (#25709; Bloomington).

### Fly images

The images were taken on Axio Zoom V16 (Zeiss) and were processed in Adobe Photoshop or Adobe Illustrator.

### CRISPR–Cas9–induced recombination on the fourth chromosome (more detailed description).

To determine if we could also induce CRISPR/Cas9–mediated recombination on the fourth chromosome, we first generated flies harboring either a *CIGAR^eGFP^* or *CIGAR^mCherry^* construct on the fourth chromosome at position *102F* ([51]). These animals are referred to as *CIGAR^eGFP,102F, w+^* and *CIGAR^mCherry,102F, w+^*, respectively. Activation of the reporters was achieved with the previously used tRNA spaced *sgRNA-1,2* tandem array (*U6:3-sgRNA^CIGAR(1,2)^*). To test if CRISPR-Cas9–mediated DSBs also result in recombination, the fourth chromosome, *nos-Cas9/Y; U6:3-sgRNA^CIGAR(1,2)^/+; CIGAR^eGFP,102F, w+^/CIGAR^mCherry,102F,w+^* G0 males were crossed to yw females (or *nosCas9/+; U6:3-sgRNA^CIGAR(1,2)^/+; CIGAR^eGFP,102F, w+^/CIGAR^mCherry,102F, w+^* G0 females to *yw* males). The offspring of such crosses were first scored at the larval stage to identify animals with either an activated *CIGAR^eGFP,102F^* or *CIGAR^mCherry,102F^* reporter. This preselection was made to ascertain that DSBs occurred in both constructs enhancing the likelihood of detecting recombination events on the fourth chromosome in case they occur. 17 vials (crosses from 12 G0 males and 5 G0 females) containing GFP and mCherry-positive larvae were selected for further analysis. A total of 172 *y,w+* F1 animals harboring either a *CIGAR^eGFP,102F^* or *CIGAR^mCherry,102F^* reporter but lacking the *sgRNA* plasmid *U6:pCFD5* (to avoid mosaic flies) from 13 of these crosses were randomly picked right after hatching. Single fly PCR of the target as well as part of the fluorophore region for the CIGAR reporters was performed for these 172 animals (Tables S2 and S3). PCR products and readable sequences were obtained in 156 cases.

For the recombination experiments using the CIGAR transgenes on chromosome 4, we estimate the recombination frequency to be 26%: 41 recombinants identified/156 flies analyzed × 100 (see Fig 2C for numbers).

### CRISPR-induced recombination between two phenotypic markers (TR in Fig 3)

To confirm that indeed recombination between sister chromatids occurs after CRISPR–Cas9–induced DSBs, we tested if we could induce recombination between two visible markers that are separated by about 100 kb (Figs 3A and S3). The markers were *w+*-marked *CIGAR^mCherry,102F, w+^* at 102F and the recessive viable mutation *sv^spa-pol^* (a mutation in the eye-specific enhancer of the *Drosophila Pax 2* gene) located downstream of the *w+* marker near the tip of chromosome 4 ([65]). The Cas9 target site was selected in the 3′UTR of the *toy* gene residing about 18 kb downstream of the *CIGAR^mCherry,102F, w+^* transgene insertion site (Fig S3). The rough-eye phenotype of *sv^spa-pol^* is reliably scored (100% penetrance) and is visible if the mutation is homozygous or over a null allele of *sv* such as *sv^Δ122^* (Sabarinadh Chilaka, Michael Daube, Erich Frei, and Markus Noll, in preparation).

*yw; CIGAR^mCherry,102F, w+^/Dp(1;4)1021,y+, sv^spa-pol^* embryos were injected with recombinant Cas9 protein complexed with in vitro–translated *sgRNA-3* targeting the 3′UTR of the *toy* gene. From the eclosing G0 animals, a total of 135 crosses were set up. Either five G0 females (29 crosses) or single G0 males (106 crosses) were crossed with homozygous *Dp(1;4)1021,y+, sv^spa-pol^* flies. F1 offspring were either *yw; CIGAR^mCherry,102F, w+^/Dp(1;4)1021,y+, sv^spa-pol^* (UR-A; phenotypically *w+, y+ sv+*) or homozygous *Dp(1;4)1021,y+, sv^spa-pol^* animals (UR-B; phenotypically *w, y+, sv^spa-pol^*) (Fig 3E). Recombination events would be phenotypically distinct (Fig 3E; TR-A or TR-B): TR-A is *yw; CIGAR^mCherry,102F, w+^, sv^spa-pol^/Dp(1;4)1021,y+, sv^spa-pol^* and would be scored as having rough, red eyes (*w+, y+ sv^spa-pol^*). The second recombination event (TR-B) that could occur would be *yw; Dp(1;4)1021,y+, sv+/Dp(1;4)1021,y+, sv^spa-pol^* flies having white and smooth eyes (*w, y+ sv+*). From a total of 8,604 scored F1 animals, 253 putative recombinant flies were recovered: 216 had rough red eyes (2.5%, from 56 independent G0 crosses) and 37 animals had white and smooth eyes (five independent G0 crosses). Notably, in one cross, 31 *w⁻, y+ sv+* animals were present in one tube corresponding to about 25–30% of the offspring. 70 of the 253 putative recombinants (we picked mostly males) were backcrossed to homozygous *Dp(1;4)1021,y+, sv^spa-pol^* flies to establish stocks. 61 of the 70 putative recombinants were recovered from independent crosses (56 *w+, sv^spa-pol^* and 5 *w, y+, sv+*). 13 of the 70 crosses remained without offspring. Most importantly, from 21 putative recombinants, we could generate homozygous viable lines. 17 of the 21 lines showed the *y, w+, sv^spa-pol^* phenotype, whereas four lines were *w, y+, sv+* marking all 21 lines as true recombinants. The remaining 36 putative recombinants were homozygous lethal (for more information on these 36 lines see "Cas9-induced DSBs may lead to loss of chromosomal structures" below).

We then investigated these animals by PCR specific for the CRISPR target site. Sequence analysis revealed that with the exception of two animals all CRISPR sites were without any indel (i.e., had a wild-type sequence; Tables S4 and S5). Interestingly, two recombinants from the same cross (G0 male) showed the same CRISPR mark (6-bp deletion). However, they had the complementary phenotype: Whereas one fly showed the *y, w+, sv^spa-pol^*

phenotype, the other was phenotypically *w*, *y⁺*, *sv⁺*, indicating that we may have recovered both chromosomes from the same recombination event. A third recombined animal from the same cross showed no CRISPR mark and thus can be counted as independent recombination event.

### Recombination experiment between two phenotypic markers (NSR in Fig 4)

To assess if CRISPR–Cas9–induced DSBs in general would enhance the frequency of recombination away from the Cas9 target site, we repeated the experiment and assessed the recombination frequency between a $y^+$ marker on the short, left arm of chromosome 4 and the $w^+$-marked *CIGAR^mCherry,102F* inserted at *102F* (see NSR in Fig 4C). To use the same experimental conditions and to exclude that the addition of active Cas9 would also induce recombination elsewhere on the fourth chromosome, embryos of the genotype (*yw; CIGAR^mCherry,102F,w+/Dp(1;4)1021,y+, sv^spa-pol*) were injected with Cas9 protein complexed with in vitro–translated *sgRNA-3*. This again would induce DSBs *3′* of *toy* located distal of *CIGAR^mCherry,102F* but should not have any influence on the recombination between the $y^+$ and the $w^+$ markers. G0 animals were this time crossed with yw animals to be able to score crossover events between the $y^+$ and the $w^+$ marker (Fig 4D). As expected, F1 animals were phenotypically either *yw⁺* or *y⁺w* because the two markers normally segregate. However, 11 phenotypically $y^+,w^+$ flies were recovered from a total of 13,220 analyzed offspring. To assess if the $y^+,w^+$ recovered animals were indeed spontaneous recombinants, or represented non-disjunction events (i.e., triplo-4 animals which are viable), we backcrossed such animals again against yw flies. Spontaneous recombination between $y^+$ and $w^+$ could be excluded as the above-mentioned backcross would have only revealed phenotypically $y^+w^+$ or yw flies. Instead, nondisjunction could be confirmed in all 11 cases because the backcross revealed phenotypically $y^+w^+$, yw, $y^+w$, and $yw^+$ flies. We determined the non-disjunction rate for the fourth chromosome to be 1 in 1,200, which is similar to the one observed for the X chromosome (66).

## Supplementary Information

## Acknowledgements

We thank Julia Compart, Angelika Dannert, Madlen Mueller, and Andri Lansel for technical help, Giuseppe Saccone, Daniel Bopp, Alessandro A Sartori, Damian Brunner, Darren Gilmour, Hugo Stocker, Olivier Urwyler, Ernst Hafen, Hirokazu Okada, Markus Noll, Christian Lehner, and Erich Frei for scientific discussions; Fillip Port for reagents; Johannes Bischoff for the *ZH attP 5D* line; Markus Noll for the *yw; sv^A122/Ci^D, sv^spa-pol* fly stock; and Martin Jinek and Giuseppe Saccone for recombinant Cas9 protein for testing. This work was supported by the Swiss National Science Foundation (SNF), grant number 310030B_173331 (K Basler), Swiss National Science Foundation, grant number 170623 (C Mosimann), Swiss Bridge Foundation (A Burger), SNF consecutive grants 31003A_162557 and 31003A_182532, and by the SystemsX.ch grant 51RT-0_145725 (to Ernst Hafen; ETH Zurich).

## Author Contributions

E Brunner: conceptualization, data curation, formal analysis, supervision, validation, investigation, visualization, methodology, project administration, and writing—original draft, review, and editing.
R Yagi: conceptualization, resources, formal analysis, validation, investigation, visualization, methodology, and writing—original draft, review, and editing.
M Debrunner: validation, investigation, methodology, and writing—review and editing.
D Beck-Schneider: investigation, methodology, and writing—original draft, review, and editing.
A Burger: resources, methodology, and writing—review and editing.
E Escher: investigation and methodology.
C Mosimann: resources, funding acquisition, methodology, and writing—review and editing.
G Hausmann: funding acquisition and writing—review and editing.
K Basler: conceptualization, resources, supervision, funding acquisition, project administration, and writing—review and editing.

### Conflict of Interest Statement

The authors declare that they have no conflict of interest.

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
