## [Reviewer comments · Life Science Alliance]

CRISPR-Induced Double-Strand Breaks Trigger Genetic Exchanges Between Homologous Chromosome Arms

Erich Brunner, Ryohei Yagi, Marc Debrunner, Dezirae Beck-Schneider, Alexa Burger, Eliane Escher, Christian Mosimann, George Hausmann, Konrad Basler
DOI: 10.26508/lsa.201800267

Review timeline:

Submission Date:	6 December 2018
Editorial Decision:	5 February 2019
Revision Received:	14 May 2019
Editorial Decision:	28 May 2019
Revision Received:	29 May 2019
Accepted:	29 May 2019

Report:

(Note: Letters and reports are not edited. The original formatting of letters and referee reports may not be reflected in this compilation.)

No Peer Review Process File is available with this article, as the authors have chosen not to make the review process public in this case.

1st Editorial Decision

5 February 2019

Thank you for submitting your manuscript entitled "CRISPR Induced Double Strand Breaks Result in an Unexpectedly High Rate of Chromosomal Recombination" to Life Science Alliance. The manuscript was assessed by two expert reviewers, whose comments are appended to this letter. Please excuse the delay in getting back to you. We had to give the reviewers more time because of the recent holidays, and we were hoping for a third report on your work that was promised, but never delivered. We therefore decided to move forward with the reports already at hand.

As you will see, the reviewers appreciate your work and provide constructive input on how to further strengthen it. We would thus like to invite you to provide a revised version of your work, addressing the issues raised by the reviewers. We would be happy to discuss the individual revision points further with you should this be helpful.

Thank you for this interesting contribution to Life Science Alliance. We are looking forward to receiving your revised manuscript.

2nd Editorial Decision

28 May 2019

Thank you for submitting your revised manuscript entitled "CRISPR-Induced Double-Strand Breaks Trigger Genetic Exchanges Between Homologous Chromosome Arms". As you will see, the reviewers appreciate the introduced changes and we would thus be happy to publish your paper in Life Science Alliance pending final revisions necessary to meet our formatting guidelines:

- Please list 10 authors et al in the reference list
- We display S figures in-line in the HTML version of the paper; Please upload the S figures as individual files and move the figure legends into the main manuscript file
- Please provide the S Tables as word docx files
- Please add a callout to Fig2A in the manuscript text

3rd Editorial Decision

29 May 2019

Thank you for submitting your Research Article entitled "CRISPR-Induced Double-Strand Breaks Trigger Genetic Exchanges Between Homologous Chromosome Arms". It is a pleasure to let you know that your manuscript is now accepted for publication in Life Science Alliance.

Congratulations on this interesting work.